# Tropical Red Macroalgae Cultivation with a Focus on Compositional Analysis

**DOI:** 10.3390/plants12203524

**Published:** 2023-10-10

**Authors:** Simona Augyte, Neil A. Sims, Keelee Martin, Stefanie Van Wychen, Bonnie Panczak, Hannah Alt, Robert Nelson, Lieve M. L. Laurens

**Affiliations:** 1Ocean Era, Inc., Kailua-Kona, HI 96740, USA; neil@ocean-era.com (N.A.S.); keeleejm@hawaii.edu (K.M.); 2Bioenergy Science and Technology Directorate, National Renewable Energy Laboratory, Golden, CO 80401, USA; stefanie.vanwychen@nrel.gov (S.V.W.); bonnie.panczak@nrel.gov (B.P.); hannah.alt@nrel.gov (H.A.); robert.nelson@nrel.gov (R.N.); lieve.laurens@nrel.gov (L.M.L.L.)

**Keywords:** aquaculture, biorefinery, carbon capture, compositional analysis, feedstock, macroalgae, polysaccharides

## Abstract

To create carbon efficient sources of bioenergy feedstocks and feedstuff for aquaculture and terrestrial livestock, it is critical to develop and commercialize the most efficient seaweed cultivation approach with a sustainable nutrient input supply. Here, we present data for a novel, onshore tropical macroalgae cultivation system, based on influent deep seawater as the nutrient and carbon sources. Two red algal species were selected, *Agardhiella subulata* and *Halymenia hawaiiana,* as the basis for growth optimization. Highest productivity in small-scale cultivation was demonstrated with *A. subulata* in the 10% deep seawater (64.7 µg N L^−1^) treatment, growing at up to 26% specific growth rate day^−1^ with highest yields observed at 247.5 g m^−2^ day^−1^ fresh weight. The highest yields for *H. hawaiiana* were measured with the addition of 10% deep seawater up to 8.8% specific growth rate day^−1^ and yields at 63.3 g fresh weight m^−2^ day^−1^ equivalent. Biomass should be culled weekly or biweekly to avoid density limitations, which likely contributed to a decrease in SGR over time. With a measured 30–40% carbon content of the ash-free dry weight (20–30% of the dry weight) biomass, this translates to an almost 1:1 CO_2_ capture to biomass ratio. The compositional fingerprint of the high carbohydrate content of both *Agardhiella* and *Halymenia* makes for an attractive feedstock for downstream biorefinery applications. By focusing on scaling and optimizing seaweed farming technologies for large-scale onshore farms, the opportunities for yield potential, adaptability to cultivation conditions, and meeting global sustainability goals through novel, carbon-negative biomass sources such as seaweed can be realized.

## 1. Introduction

As global population continues to rise, the aquaculture sector is becoming more critical for the provisioning of food while delivering economic and social profits [1,2,3,4,5]. Currently, macroalgal farming is gaining attention as a source of climate, environmental, and socioeconomic solutions, ranging from carbon dioxide mitigation and nutrient filtration to being a source of renewable feedstock materials for food, feed, and biofuel [6,7,8]. Global production values of 32.4 million metric tonnes fresh weight (MMT FW) were reported for 2018, most of which are produced for human food [9]. Assuming the long-term growth rate (from 2000–2018) of 6.2 yr^−1^, global seaweed aquaculture is expected to reach production rates of 252 MMT FW by 2050 [10]. The commercial value of macroalgae depends largely on applications such as food, nutraceuticals, and pharmaceuticals as they contain various bioactive substances like polysaccharides, proteins, lipids, and polyphenols with various properties including anticarcinogenic, antiviral, anti-inflammatory, antioxidant, and anticoagulant activities [7,8]. Addressing critical improvements in biomass, bioproduct, and biofuel productivity is a priority for both a microalgae- and macroalgae-based bioeconomy [9,10]. A marine sustainable bioeconomy will rely on effective deconstruction of macroalgal biomass in order to utilize all valuable constituents in an economically feasible, net-negative carbon, cascading process with zero waste [11]. The term green-engineered seaweed production refers to the restorative and regenerative process of seaweed farming that is a nature-based solution to environmental degradation and resource over-exploitation, including the ongoing and historic anthropogenic disturbances of climate balance and nutrient flows [12,13]. Large scale production of macroalgae can lead to a drawdown of CO_2_ from the atmosphere [10,14,15] and has been proposed as one of the more efficient, low-energy-input, low-carbon-intensity food production systems [16] providing a host of ecosystem services (e.g., nutrient bioextraction and mitigation) and livelihoods to the coastal environment, including oxygenation of the water column, pollution bioremediation, enhancing fisheries, and reducing localized acidification [17,18,19,20,21].

Seaweed farming has developed as an alternative to exploiting natural marine ecosystems for feedstuff [22]. Land-based seaweed cultivation offers advantages in production of high value species over ocean-based farming with tighter control of biotic factors, including herbivory and disease, and abiotic factors, including optimized light, nutrient availability, seawater inputs, and aeration [23,24,25]. As biomass quality and productivity are highly dependent on the growing environment, seaweed farmers can reduce variability and obtain higher quantity and quality of targeted compounds in land-based systems, which is also important for product traceability and safety [22,23].

Macroalgae possess a myriad of different bioactive molecules and thus have great potential for development as products in the nutraceutical, functional food, and pharmaceutical markets, in addition to the current uses in industrial, agricultural, food, feed, and nutraceutical applications [26]. The bioactive properties in macroalgae polysaccharides include antioxidant, antibacterial, antiviral, anticarcinogenic, anticoagulant, and others [27]. Additional applications for the plethora of polysaccharides in macroalgae are still waiting to be discovered, and it is therefore imperative to better understand the compositional analysis of macroalgae with high yield potential in aquaculture systems.

The variability in macroalgal production values is substantial and seasonally dependent. For example, *Gracilaria chilensis* grown on suspended cultures near salmon cages in Chile showed summer daily mean growth rates of 4% with mean biomass production of 1600 g FW (fresh weight) m^−1^ month or 53 g WW m^−1^ day^−1^, while in the fall, productivity rates dropped by 40% [28]. Line trials in the tropical waters of northeast Brazil with *Gracilaria birdiae* gave yields of 1900 g FW m^−1^ with max specific growth rate (SGR) of 7.45% day^−1^ and a mean of 4.4% day^−1^ [29]. An intensive tank cultivation study with *G. chilensis* reached summer maximum biomass production of 100 g FW m^−2^ day^−1^ or an average of 4.1 kg dw m^−2^ year^−1^, equivalent to 4100 dry metric tonnes (DMT) km^−2^ yr^−1^ [30]. In Costa Rica, reported extrapolated yields of cultivated tropical *Gracilaria cervicornis* reach 76 DMT m^−2^ yr^−1^ [31]. Meanwhile, the tetrasporophyte *Asparagopsis armata* reached maximum production values in winter of 71 g DW (dry weight) m^−2^ day^−1^, while in the early summer months of up to 125 g DW m^−2^ day^−1^ [32]. This would correspond, if scalable, to between 25,915 and 45,625 DMT km^−2^ yr^−1^ but only if such high productivities can be maintained.

In addition to variation in yields, the biochemical composition of the biomass can vary seasonally even in land-based culture systems [33,34] but potentially less in tropical environments with lower seasonal climatic fluctuations. The seaweed compositional profile (i.e., the respective protein, carbohydrate, and lipid content of the harvested biomass) is important to predict and plan the route the biomass can take for either conversion or towards applications in the aquaculture sector. We have produced a compilation of literature-reported values of biochemical, specifically polysaccharide, composition of more common seaweed genera. The data in Figure 1 shows an overview of documented carbohydrate content and polymer structures (e.g., ulvan, laminarin, fucoidan, alginic acid, etc.) as reported on a dry weight basis in the literature reports. The data illustrate diversity of polysaccharides typically found in seaweeds, at concentrations that in almost all cases make up the majority of the dry biomass and are phylogenetically distinct between each of the major categories of green, brown, and red seaweeds. The brown seaweeds are dominated by combinations of laminarin, fucoidan, and alginic acid at varying proportions; the green seaweed polysaccharides are dominated by glucuronans, and in *Ulva*, the ulvan and the red seaweeds contain perhaps the most diverse polymers such as xylan, cellulose, and starch, in addition to carrageenan or agar and porphyran.

High biomass productivity per unit surface area shows potential for land-based cultivation. It is in this context that we aimed to demonstrate the maximum growth rate potential of several tropical macroalgae in Hawai’i in a land-based system following a controlled supply of nutrients from deep cold (4–8 °C) seawater, pumped up from 900 m depths at the Natural Energy Laboratory of Hawai’i Authority (NELHA) and mixed with surface seawater. Two species of red macroalgae (Rhodophyta), *Agardhiella subulata* (C. Agardh) Kraft and M. J. Wynne and *Halymenia hawaiiana* Hernández-Kantún and A. R. Sherwood, were chosen for the trials in this study due to their range of desirable qualities. These particular species were selected because they have been a part of the system for several years and have demonstrated exceptional adaptability. They have undergone vegetative propagation for multiple seasons and have become firmly established within the culture system, consistently delivering high year-round performance.

The first species used in the trials, *A. subulata*, is widely distributed along the coast of the western Atlantic, including North America, the Caribbean, South America, and the Gulf of Mexico, is a model marine species for biomedical research used in the aquarium trade, and is wild harvested for use in food grade phycocolloids [35,36]. According to records, cultivation of this species has been conducted at NELHA since 1991, with stringent biosecurity controls in place to prevent unintentional release into the wild [36]. The second species, *Halymenia hawaiiana*, was collected from Mahai’ula Bay, Hawai’i Island, vegetatively propagated for several years, and species identity confirmed based on published genetic sequences [37]. This macroalga holds significant commercial potential due to its culinary applications among diverse ethnic groups in the Hawaiian Islands, each preparing it in distinct ways [38].

The cultivation of macroalgae in tropical climates is primarily driven by the carrageenophyte industry, with the main focus on the red macroalgae, *Kappaphycus* spp., and *Eucheuma* spp. [39], while other species have been reported on with experimental trials only. Our study places emphasis on two lesser-known species with promising commercial potential. Our objectives included conducting a comprehensive analysis of the proximal composition and fatty acid profiles of the cultivated *A. subulata* and *H. hawaiiana*. This research sought to determine the polysaccharide content in their dry tissue along with other key molecules and their specific carbohydrate compositions based on variations in nutrient additions.

The outcomes of this innovative research can serve as a valuable resource for the establishment of both aquaculture facilities, enabling the scalable production of marine biomass, and the further development of seaweed value-chain products with essential biomolecules. Specifically, for this paper, our goal was to (1) give an overview of demonstrated productivity for two commercially important tropical macroalgal species, *A. subulata* and *H. hawaiiana,* when grown in an innovative land-based system; (2) extrapolate the production rate potential to large-scale intensive tropical macroalgae cultivation; and (3) compare tissue composition of carbon and nitrogen for sampling locations in a tank system with varying nutrient concentrations.

## 2. Results

### 2.1. Trial 1 Agardhiella subulata

Growth for all three treatments followed a power curve, starting out with high SGRs but slowing down over time. Both the 5% and 10% DSW treatments concluded week 5 with similar masses (5111.4 ± 393.6 g and 5279.2 ± 1744.1 g, respectively) (Figure 2). There was an effect of the DSW treatment and a clear effect of regression across weeks (*p* < 0.001, Table 1). Furthermore, the 1% DSW treatment differed with time compared to the others (*p* < 0.001). But, this analysis also suggests that biomass patterns between DSW treatments differ between periods (*p* = 0.044) and that the slopes of lines of biomass versus weeks differ between periods (*p* < 0.001).

SGR was analyzed at weekly intervals for each DSW treatment. Across all nutrient addition treatments, SGR decreased from week 1 of the trial at 24–26% SGR to week 5 at 6–7% SGR. After the trimming event in week 6, SGR increased again to a range of 15–23% and dropped to 9–12% in week 9 (Figure 3). This trial shows a similar trend to previous trials, where SGR decreases as algal biomass increases, perhaps as tank space becomes more limited. Overall, highest production values were observed in the 10% DSW treatment at 247.5 g m^−2^ day^−1^ fresh weight, or approximately 25 g m^−2^ day^−1^ dry weight for the 9-week cultivation window reported (Table 2). In the 1% DSW treatment, the pigmentation loss was observed after week two, while there was no pigmentation loss in either of the 5% and 10% DSW treatments throughout the duration of the trial (Figure 4).

### 2.2. Trial 2 Agardhiella subulata

The results in the second week of the trial revealed the highest SGRs in the two tanks directly supplied with 5% DSW at 13.2% (tank 5) and 13.52% (tank 9) (Figure 5). As nutrients trickled from the first series of tanks to subsequent ones, SGR decreased to 11.67% (tank 5), and finally to 8.38% (tank 9). The tanks supplied with only SSW had slightly higher SGR closer to the seawater inputs at 4.27% in tank 1, followed by 1.40% in tank 2 and 1.08% in tank 3. A similar trend was observed in the third tank series where tank 7 had an SGR of 3.75%, while the subsequent tank 8 had an SGR of 1.00%. From a starting biomass of 300 g, after two weeks of cultivation, the highest weights measured were in tanks 4 and 9 at 1903 g and 1991 g, respectively.

### 2.3. Trial 3 Halymenia hawaiiana

For *H. hawaiiana*, the highest overall SGRs per treatment were the cascade tanks receiving the 10% DSW additions at 6.1% (tank 4), 6.4% (tank 5), and 5.1% (tank 6) in the second series (Figure 6). The highest yields were measured in tank 5 from weeks three to four with 821 g biomass increase tank^−1^ week^−1^ for an average SGR of 8.8% day^−1^. This equates to yields of 63.3 g FW m^−2^ day^−1^ equivalent or 6.3 g m^−2^ day^−1^ dry weight at a stocking density of 900 g (Table 3).

Bleaching of *Halymenia* thalli was observed after the second week in the treatments with no DSW inputs (Figure 7), while pigment change was not observed in treatments with DSW additions.

### 2.4. Biomass Composition Results

For both *Halymenia* and *Agardhiella* samples, proximate composition was determined as ash, lipid, protein, and uronic acid composition with carbohydrate monosaccharide profile and lipid fatty acid profiles included (Table 4, Table 5, Table 6, Table 7, Table 8 and Table 9). There are some proximate compositional changes that can be observed from the trends that follow the nutrient cascade Trial #3; a higher concentration of nitrogen results immediately in a higher protein content (up to almost 16% of the dry biomass), and the depletion of nitrogen increases the measured total carbohydrate content in the biomass from 23% to over 40% of the dry biomass for *Halymenia* or 60.2% on an ash-free basis (tank 3). Similarly, for Trial #1, *Agardhiella*, the carbohydrate content almost doubles upon nutrient depletion to just over 20% of the dry biomass or 44.5% on an ash-free basis (tank 7). The higher carbohydrate containing samples are also the samples with the lowest protein content, which is consistent with the reduced productivity observed in those tanks.

We compared total carbon and nitrogen content for the *Halymenia* and *Agardhiella* biomass harvested under several nutrient physiological conditions and noticed a close to two-fold difference in respective carbon content on a dry weight basis (Table 4 and Table 7). We observed a fairly narrow range of carbon (C) content on a dry weight basis of 27.1 ± 0.8% for *Halymenia* and 18.6 ± 1.5% for *Agardhiella* biomass. Assuming all carbon is organic, then the C content of the ash-free (AF) biomass is closer between the two species (between 42 and 46% AFDW for *Agardhiella* and between 39 and 41% AFDW for *Halymenia*). The highest C content of the biomass was observed for both species in the samples associated with the highest nutrient depletion (with associated DSW inputs of 0 or 1%).

## 3. Discussion

Our pilot land-based grow-out trials demonstrate the potential for large-scale production of *Agardhiella subulata* and *Halymenia hawaiiana* for feedstuff. Major findings indicate that nutrient additions had a significant effect on biomass production over time. *Agardhiella subulata* doubled its biomass when DSW additions were increased from 1% to 5%, while still maintaining its pigmentation, indicating it was no longer nitrogen-limited. Furthermore, the trimming event in the middle of Trial #1 confirmed that when the space limitation is removed and adequate nutrients are available, the macroalgae can reach growth rates of up to 26%. A rapid decrease in SGR was observed as biomass increased in the tank over time; therefore, developing a consistent harvesting strategy is recommended to avoid light, nutrient, and density limitations. In a similar study conducted with *A. subulata* in an integrated multi-trophic aquaculture system (IMTA), the macroalga grew up to 14.4% day^−1^ and experienced density-limited growth due to self-shading and crowding, reaching its carrying capacity at 17–20 kg m^−3^ even when nutrients were still abundantly available [35]. In our study, *A. subulata* yields peaked at 247.5 g FW m^−2^ day^−1^ in the 10% DSW treatment with an average stocking density of 3.1 kg m^−2^.

Primary productivity is typically limited in tropical, oligotrophic environments by low nutrient levels in coastal waters [40,41]. Therefore, for successful cultivation of macroalgae in such nutrient-replete environments, some form of external inputs is usually required. For example, a Hawaiian community-based operation with the red *Gracilaria parvispora* grown on the island of Moloka’i was established to develop sustainable, integrated culture system with fish and shrimp effluent nutrient loading prior to placement in a low-nutrient lagoon [42]. Initial trials with sporelings from the hatchery showed low growth rates in cages over 52 weeks at 2.64% day^−1^ with yields of 14.8 g FW m^−2^ day^−1^ [39]. A follow-up study with a seven-day pulse fertilization on land prior to outplanting in floating cages for 14 days measured SGR of up to 10% with production of thalli in cages of 39–57 g DW m^−2^ day^−1^ over a 21-day production cycle [43]. Our cascade trials show similar trends of spikes in growth with higher nutrient inputs. For example, in Trial #2, additions of 5% DSW showed SGRs of up to 13.5% and maximum yields of 130 g FW m^−2^ day^−1^. Meanwhile, in Trial #3, *Halymenia hawaiiana* maximum % SGR peaked at 6.4% with maximum yields in 10% DSW in week four at 63 g FW m^−2^ day^−1^. Overall, *A. subulata* at 10% DSW outperformed *H. hawaiiana* at the same nutrient concentration. While temperature was not monitored throughout the entire system, it is possible that slight decreases, attributed to the cooler DSW additions, may have influenced overall productivity.

Through species selection and cultivation optimization, a land-based cultivation system could be developed to produce macroalgal biomass at rates far exceeding terrestrial crop production. With a measured 20–30% carbon content of the dry weight biomass (or 30–40% on an ash-free basis) for both species tested, the production translates to an almost 1:1 CO_2_ capture to biomass ratio; large-scale deployment opportunities exist for carbon capture from naturally dissolved inorganic carbon in seawater. The very high ash content measured in *Agardhiella* (50–60%) could be associated with inorganic carbonates (as calcium carbonates) that are present at a much higher concentration with *Agardhiella* biomass compared to other tropical seaweed species, but this remains to be characterized. Only the measurable carbohydrates are reported and summed and may be an underestimation of the intact complex polysaccharides present in the biomass. The overall measured lipid content, as fatty acid methyl esters, does not change with the changes in nutrients for either species and remains very low (<2% on an ash-free basis), which is consistent with the role of lipids in macroalgae as structural, e.g., membrane glyco-lipids, rather than energy-storage functions [40].

This cultivation system presents an opportunity for generating a carbon-neutral crop with significant organic and inorganic carbon capture potential. Based on the above-discussed large-scale deployment, the intensive land-based cultivation can create a 18–20 tonnes (T) acre^−1^ direct carbon capture potential, far exceeding that of terrestrial net ecosystem productivities [44]. Typical terrestrial (conventional) corn farming yields approximately 4.5 T acre^−1^ yr^−1^ (or just over 1.1 T m^−2^ yr^−1^). The compositional fingerprint at high carbohydrate content of *Agardhiella* makes for an attractive feedstock for downstream conversion to bioenergy (e.g., fermentation to ethanol) and feedstuff. The measured monosaccharide profile of the detected carbohydrates in *Halymenia* (at over 60% of the ash-free organics) indicates easily fermentable sugars dominate the profile (e.g., glucose and galactose), and thus a path to ethanol or other, ideally more carbon-efficient, fermentation-derived products may be feasible. With a sustained effort on scaling and deploying seaweed farming technologies to large-scale onshore and ultimately offshore farms, economic opportunities can be achieved for underrepresented communities. In particular, it has been noted that to meet global sustainability goals, novel, carbon-negative biomass sources such as these attractive seaweed sources described in this study will become needed solutions [10].

The higher carbohydrate-containing samples, for both species, are also the samples with the lowest protein content. This is consistent with the reduced productivity observed in those tanks, indicating that the nutrient innovation based on deep seawater delivery can be used to tune the composition of the biomass for further applications. The selection of macroalgal species plays a crucial role in determining the success of product development. It has been suggested to prioritize cultivation of the highly nutritious native Hawaiian macroalgae, including *Halymenia formosa* (protein content up to 21.2%), *Porphyra vietnamensis* (with 6.4 times more β-carotene than reported for spinach at 67 IU g^−1^), and *Monostroma oxyspremum* (with a caloric content of over 3000 cal g^−1^ ash free dry weight) [45]. Furthermore, it is equally important to consider factors such as yield potential and adaptability to cultivation conditions to ensure economic viability for farmers.

## 4. Materials and Methods

### 4.1. Land-Based Macroalgae Cultivation in Onshore Tanks

For the onshore cultivation and growth experiments at Ocean Era, Inc., at the NELHA facility, outdoor tanks were set up directly connected to a seawater supply system (Figure 2). The 340 L capacity tanks were filled to 300 L with flow-rates at 50 turnovers/day. The seawater was continuously supplied from pipes connected to surface seawater (SSW) from the surface and deep-seawater (DSW) pumped from pipes at 900 m below the surface, with nutrient composition of seawater continuously tested at the NELHA Water Quality Lab. The SSW is low in available nitrogen at 7.4 µg N L^−1^, and the DSW has a high nutrient profile of 580.5 µg N L^−1^ (including NO_3_^−^, NO_2_^−^, NH_4_^+^, and NH_3_), and both can be supplemented to the tanks at various concentrations depending on experimental design. For these trials, 10% addition of DSW was added with the combined nitrogen available at 64.7µg N L^−1^, a 5% addition of DSW at 36.1 µg N L^−1^, a 1% addition of DSW at 13.1 µg N L^−1^, and the 0% DSW contained 7.39 µg N L^−1^. Photosynthetically active radiation (PAR) was measured by installing HOBO devices (Onset Computer Corporation) just under the surface of the tank. Shade cloth (75%) was used for all treatments, and max daily PAR was recorded at 400 μmol m^−2^ s^−1^.

All tanks were aerated through a single diffuser hose running along the center of the length of the tank (Figure 8). Each tank was supplied with a continuous flow-through of temperature-controlled (25.5 °C) SSW at 200 mL s^−1^ filtered to 300 µm, with an addition of unfiltered DSW at a flow rate specific to their treatment. Flow rates were hand-calibrated daily to their assigned DSW concentrations.

Here, we describe three separate trials set up for the tropical red macroalgae *Agardhiella subulata* and *Halymenia hawaiiana* at varying levels of DSW ranging from 0% to 10% to test the effects of nutrient additions on productivity. The first trial with *A. subulata* was a nine-week trial with a trimming event in the middle that brought the biomass down to the original mass. Because of the constraints in farm workflow, we were only able to conduct the initial long trial with *A. subulata*, and it was impractical to simultaneously run a trial with *H. hawaiiana.* The second and third trials, with *A. subulata* and *H. hawaiiana*, respectively, were cascading trials that were plumbed to allow water to flow successively in a row with 3 tanks per series (n = 3). The first series was a negative control and only received SSW inputs in tank 1. The second series had DSW inputs in tank 4, and the second series had DSW inputs in tank 9.

Photos were taken on weigh days to monitor the macroalgal pigment and overall condition for *A. subulata* in Trial #1 and *H. hawaiiana* for trial #3. Tanks were randomly chosen as representatives for their respective treatment types, ensuring that the same biomass was photographed each week. For each trial, biomass from each tank was weighed by manually scrubbing and clearing away any fouling on a weekly basis. This was performed by removing all biomass using a dip net, spinning off excess water using a salad spinner for 5 min, and then weighing. Fresh weight (FW, in grams) was recorded individually for each tank, and SGR % growth day^−1^ was calculated using the equation:SGR = 100 ∗ (ln(w_final_)−ln(w_initial_))/time(1)
where w = wet weight (in grams), and time (in days) is the interval between w_final_ and w_initial_.

Statistical analysis was conducted for Trial #1. All weight outcome variables were analyzed using a repeated measures ANOVA with nutrient treatment, week (time), period, and their interaction as fixed effects. Because the trimming event took place in the middle of the trial, the first 5 weeks were treated as period A and the post trimming event with the second 5 weeks as period B. The data met the parametric assumptions of normality of residuals and homogeneity of variances. The relationship between biomass and week is linear after log transforming both variables (i.e., power curve), and thus the log transformed variables were used in the analysis. All analyses were performed using R version 4.3.1 (R Core Team 2023) with the package *ez* [46].

At the end of the trials, the tissue samples were given a quick (<30 s) freshwater rinse, placed into plastic bags, and frozen for compositional analysis. Here, we present compositional analysis data for biomass from Trials #1 and #3.

### 4.2. Trial #1 Agardhiella subulata Trimming Trial

The first experimental trial with *Agardhiella subulata* was set up to test how the effects of three nutrient concentrations would affect significant growth rate (SGR) over time. Additionally, it was further investigated how the effects of trimming, thus reducing density limitation in the tank, would affect SGR.

The biomass in this trial was stocked from a holding tank held at ~9% DSW with an average temperature of 23 °C and average max PAR of 523 µmol photons/m^−2^s^−1^ for two weeks prior to the trial start. Nine identical tanks were randomly and evenly stocked with an average of 30.4 ± 0.3 g of *A. subulata* from the source tank. Three DSW treatments of 1%, 5%, and 10% concentrations were randomly assigned in triplicate with a continuous supply of DSW at that specific concentration. A “trimming” event was conducted immediately following weight data collection at the week five event, whereby the existing biomass in each tank was reduced to 30 g FW. The trial ran for a total of 9 weeks.

### 4.3. Trial #2 Agardhiella subulata Nutrient Cascade Cultivation Trials

The second experimental trial with *Agardhiella subulata* was set up with cascading tanks to test the addition of nutrients to the system and how these are taken up sequentially from one tank to the next.

The seaweed stocked in this trial were held only in SSW for two weeks prior to the trial to deplete their nutrient reserves. At the start of the trial, nine identical tanks were randomly and evenly stocked with an average of 300.3 ± 0.4 g of *Agardhiella subulata* from the source tank. The trial ran for 2 weeks. Continuous inputs of 5% DSW were added to the first tank in series 2 (tank 4) and again to the last tank in series 3 (tank 9). Series 1 only received SSW and acted as a negative control for the study.

### 4.4. Trial #3 Halymenia hawaiiana Nutrient Cascade Cultivation Trial

The third experimental trial was run with *Halymenia hawaiiana* for 4 weeks to observe biomass production. Starting biomass came from at holding tank with 10% DSW continuous inputs at 300 g for a stocking density of 1 g/L in each of the outdoor tanks. Similar to Trial #2, tanks were set up in a cascade with three tanks per series (n = 3) receiving the downstream trickle from the initial tank. In the second series, tank 4 received 10% DSW addition. The third series received 10% DSW in tank 9.

### 4.5. Macroalgae Compositional Analysis

All compositional analyses were performed on lyophilized biomass derived from Trials #1 and #3 described above, following previously documented and published procedures summarized here briefly [9,47,48,49,50]. For moisture content, an aliquot of the sample was dried at 40 °C for 2 days, and a percent moisture was determined [51]. All subsequent values were determined on a moisture-free (dry weight) basis. Dry oxidation (ashing) overnight at 575 °C was performed on the oven-dried sample to determine the ash content of the biomass [51]. For carbohydrate content, a two-step sulfuric acid hydrolysis was used to hydrolyze the polymeric forms of carbohydrates in the biomass into monomeric subunits. The monomers were then quantified by high-performance anion exchange chromatography with pulsed amperometric detection (HPAEC-PAD) [48,49]. The separation and detection method used to characterize the biomass follows a minor deviation from the published procedure; after hydrolysis, samples were filtered through 0.2 µm nylon filters. Monomeric sugars (mannitol, fucose, rhamnose, glucosamine, arabinose, galactosamine, galactose, glucose, mannose, xylose, and ribose) released during hydrolysis were injected onto and analyzed on a Dionex ICS-5000+, HPAEC-PAD system equipped with a PA-1 column (Dionex #035391, ThermoFisher, Waltham, MA, USA) and guard cartridge (Dionex #043096) set to 35 °C. The column was washed at 1 mL/min for 10 min at 200 mM NaOH then equilibrated for 30 min at 14 mM NaOH, after which a 20 min isocratic run at 14 mM NaOH separated the carbohydrates, which were detected by a pulsed amperometric detector (PAD). The uronic acids were also quantified for each of the hydrolysates using the HPAEC-PAD as described above apart from the eluents and gradient used. To separate four uronic acids (galacturonic, guluronic, glucuronic, and mannuronic), the following gradient was utilized: 10-min pre-run column flush with 200 mM NaOH, 20-min pre-run equilibration with 1 mM sodium acetate/100 mM NaOH at 17% (remainder 18.2 megaohm water), and a 10-min run with the 17% 1 mM sodium acetate/100 mM NaOH (remainder 18.2 megaohm water) [52]. Whole biomass transesterification of lipids to fatty acid methyl esters (FAME) was performed to quantify the lipid content as FAME [48]. Macro-algae-specific analyses were included, such as the analysis of additional carbohydrate monomers (e.g., 3,6-anhydrogalactose), and uronic acids were quantified on an HPAEC-PAD system using a CarboPac-PA1 column and guard as described previously [51,52].

## 5. Conclusions

This manuscript describes a novel approach for intensive onshore tropical seaweed cultivation of *Halymenia* and *Agardhiella* using an innovative nutrient delivery cultivation tank cascade based on deep seawater supplementation. The fast growth rates translate to yield and productivity rates that exceed terrestrial agriculture net ecosystem productivities and thus present a carbon-neutral biomass source, while simultaneously providing an attractive, carbohydrate-rich biomass for high-value applications. Follow-up trials are recommended to examine the effects of slight decreases in temperature coupled with nutrient inputs on growth and to assess temperature impact considering that DSW is cooler than surface seawater. Additional trials examining other highly productive tropical species with commercial potential are warranted. Carbon drawdown via macroalgae cultivation with attractive biomass composition opens opportunities for downstream conversion of high value applications. Furthermore, collaboration between the two fast-growing sectors in the blue economy of seaweed cultivation and the conservation industry can bring marine forest restoration into commercially relevant global scales to meet global sustainability targets [6,53].

## Figures and Tables

**Figure 1 plants-12-03524-f001:**
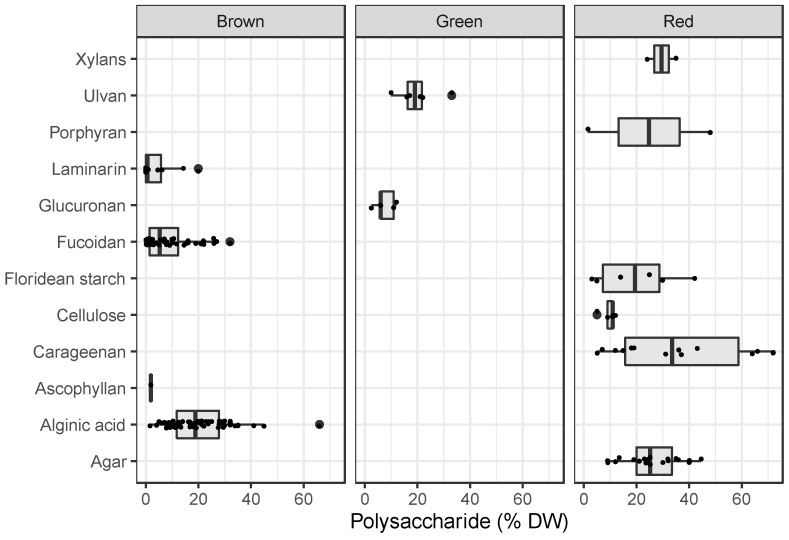
Overview of documented carbohydrate (polysaccharide) content in seaweeds in the literature, shown as reported on a dry weight basis and grouped by major category of seaweed: brown, green and red. Raw data and compilation of literature sources spreadsheet available as Appendix A.

**Figure 2 plants-12-03524-f002:**
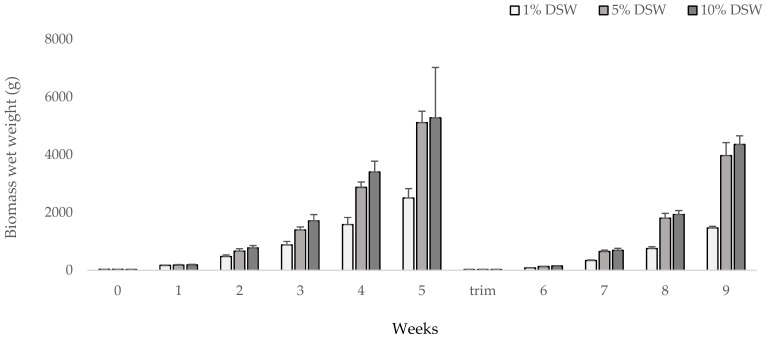
Trial #1. Tank cultivated *Agardhiella subulata* production under three different scenarios of nutrient additions (1%, 5%, and 10% DSW; tank n = 3) over 9 weeks. Standard error bars are included. A trimming event took place after the weigh-in on week 5 when biomass was brought down to the original mass of 30 g.

**Figure 3 plants-12-03524-f003:**
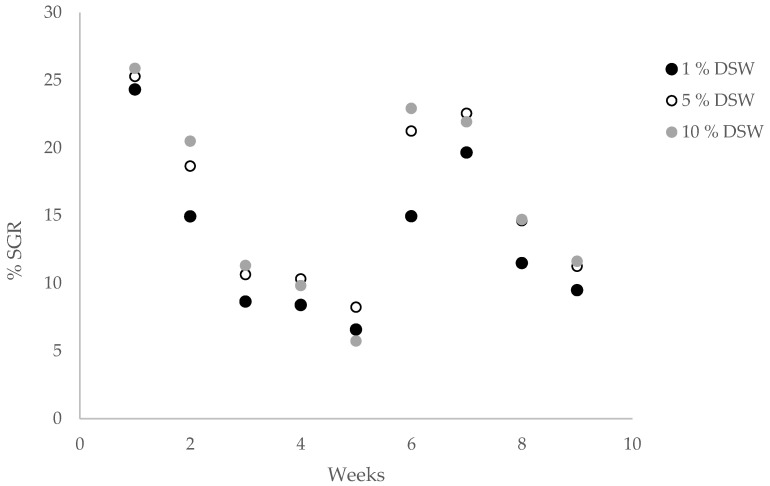
Average specific growth rate (SGR % day^−1^) and yields in g FW m^−2^ day^−1^ equivalent for *Agardhiella subulata* Trial #1 for each treatment (n = 3) with varying levels of nutrients derived from deep seawater (DSW) supplementation.

**Figure 4 plants-12-03524-f004:**
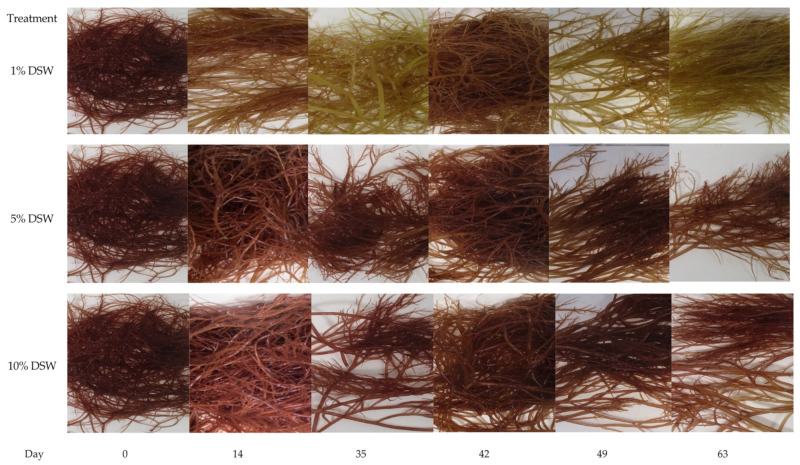
Trial #1. Pigmentation of *A. subulata* under three different nutrient regiment conditions over the 9-week trial.

**Figure 5 plants-12-03524-f005:**
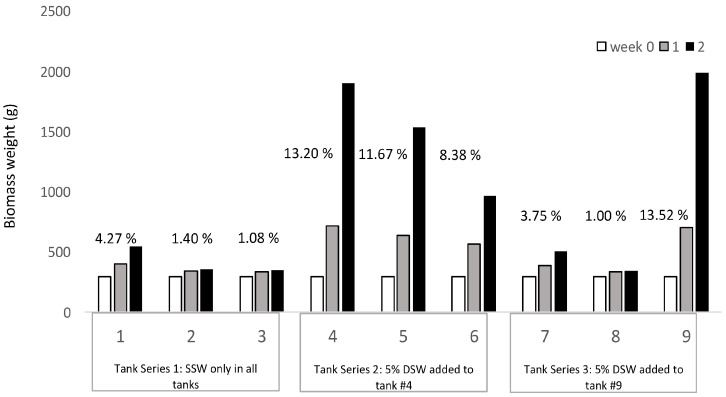
Trial #2. Tank cultivated *Agardhiella subulata* production in a cascade trial where nutrients (5% DSW addition) were only supplied to the inflow of tank 4 and trickled down subsequently to tanks 5 and 6 and nutrients were again added to tank 9. Percent SGR (% growth day^−1^ as a weekly average) listed above the bars.

**Figure 6 plants-12-03524-f006:**
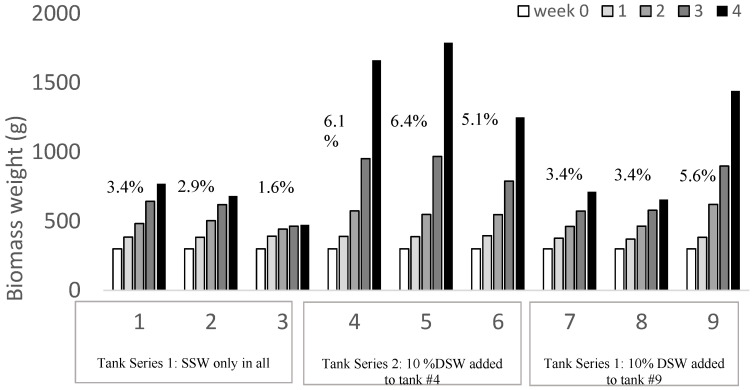
Trial #3. Tank cultivated *Halymenia hawaiiana* production over 4 weeks in the cascade trial with 10% DSW inputs into tank #4 and tank #9. Percent SGR (% growth/day as a weekly average) listed on bars.

**Figure 7 plants-12-03524-f007:**
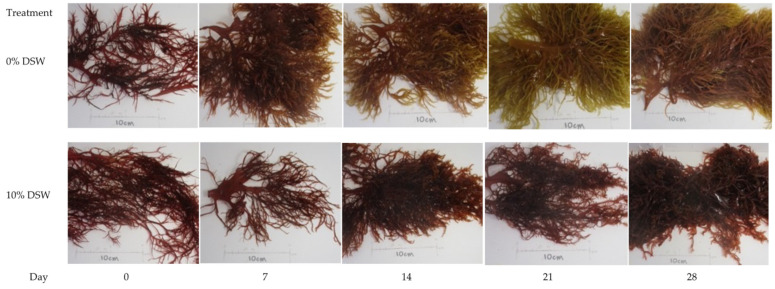
Trial #3. Pigmentation of *Halymenia hawaiiana* under different nutrient additions (0% and 10% DSW).

**Figure 8 plants-12-03524-f008:**
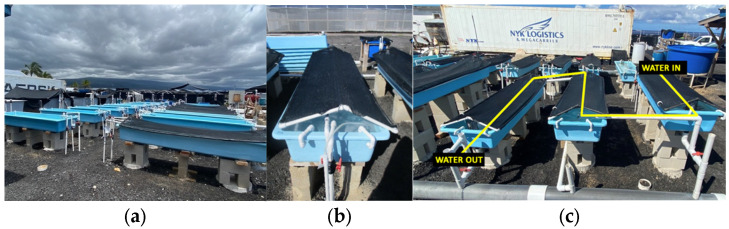
(**a**) Set up of trial at Ocean Era land-based facilities; (**b**) close up of cultivation tank; (**c**) cascade trials (#2 and #3) showing water flow through 3 tanks, each approximately 1.42 m^2^ in illuminated surface area.

**Table 1 plants-12-03524-t001:** Statistical outcomes of repeated measures ANOVA model of the effect of treatment and time and their interaction effects on SGR for Trial #1. Weeks 0–5 are considered Period A, and weeks 6–9 following trimming are considered Period B. Statistically significant *p*-values are indicated in bold.

Source	Df	DFd	F	*p*-Value
DSW treatment	2	6	145.99	**<0.0001**
Period	1	6	2.72	0.15
Week	1	6	44,379.72	**<0.0001**
DSW treatment:Period	2	6	5.75	0.044
DSW treatment:Week	2	6	189.04	**<0.0001**
Period:Week	1	6	17.63	**0.006**
DSW treatment:Period:Week	2	6	3.55	0.096

**Table 2 plants-12-03524-t002:** Average yields in g FW m^−2^ day^−1^ equivalent for *Agardhiella subulata* Trial #1 for each treatment (n = 3) with varying levels of nutrients derived from deep seawater (DSW) supplementation.

Week	1 % DSW (*v*/*v*) Yield	5% DSW (*v*/*v*) Yield	10 % DSW (*v*/*v*) Yield
1	14.0	15.3	15.8
2	31.7	49.5	60.3
3	40.8	74.6	95.7
4	71.9	50.7	172.3
5	93.7	228.4	191.6
TRIM			
6	5.6	10.5	12.1
7	25.8	52.5	55.5
8	42.6	118.0	126.6
9	72.4	221.0	247.5

**Table 3 plants-12-03524-t003:** Average yields in g FW m^−2^ day^−1^ equivalent for *Halymenia hawaiiana* cascade Trial #3 with varying levels of nutrients derived from deep seawater (DSW) supplementation at 0% or 10% of the volume (n = 2).

Week	0% DSW (*v*/*v*) Yield	10% DSW (*v*/*v*) Yield
1	8.2	8.8
2	9.2	21.3
3	13.7	33.0
4	13.4	63.3

**Table 4 plants-12-03524-t004:** Proximate biochemical compositional analysis and elemental compositional analysis of *Agardhiella subulata* (Trial #1), shown on the basis of dry biomass harvested from nutrient trials. Samples were taken at the end of the trial.

Tank #	Ash (%)	Lipid (%)	Protein (%)	Carbs. (%)	Uronic Acids (%)	Carbon (%)	Nitrogen (%)
1% DSW (Tank 1)	57.04	1.45	5.03	18.27	0.00	18.90	3.23
5% DSW (Tank 2)	61.63	1.27	7.35	12.59	0.00	17.22	2.94
10% DSW (Tank 3)	59.75	1.41	10.20	11.88	0.00	16.98	2.83
1% DSW (Tank 7)	53.82	1.25	4.41	20.56	0.00	19.68	3.39
5% DSW (Tank 8)	53.29	1.47	9.50	16.91	0.00	20.91	3.47
10% DSW (Tank 9)	60.77	1.45	10.55	11.68	0.00	18.09	3.04

**Table 5 plants-12-03524-t005:** Carbohydrate monomeric composition of *Agardhiella subulata* (Trial #1), shown on the basis of dry biomass harvested from nutrient trials. Samples were taken at the end of the trial. <loq (below limit of quantification) indicates where component peaks were detected, but the areas were below the calibration curve and thus no quantification was possible; Rham = rhamnose, GalN = galactosamine, Ara = arabinose, GlucN = glucosamine, Gal = galactose, Gluc = glucose, Man = mannose, Xyl = xylose, Rib = ribose.

Tank #	Rham	GalN	Ara	GlucN	Gal	Gluc	Man	Xyl	Rib
1% DSW (Tank 1)	<loq	<loq	<loq	<loq	8.45	2.28	0.69	0.24	0.23
5% DSW (Tank 2)	<loq	<loq	<loq	<loq	8.66	2.76	0.72	0.25	0.20
10% DSW (Tank 3)	<loq	<loq	<loq	<loq	10.98	4.42	0.83	0.44	0.24
1% DSW (Tank 7)	<loq	<loq	<loq	<loq	8.25	2.32	0.68	0.23	0.20
5% DSW (Tank 8)	<loq	<loq	<loq	<loq	11.97	7.43	0.73	0.43	<loq
10% DSW (Tank 9)	<loq	<loq	<loq	<loq	11.82	5.43	0.65	0.38	<loq

**Table 6 plants-12-03524-t006:** Fatty acid profile of *Agardhiella subulata* (Trial #1), shown on the basis of total FAME content in harvested biomass from nutrient trials. Samples were taken at the end of the trial. All fatty acids detected are noted as chain length with number of unsaturated bonds, with carbon position from terminal end indicated for first unsaturation.

Tank #	C14:0	C16:0	C16:1n7	C16:1n5	C18	C18:1n9	C18:1n7	C18:2n6	C18:3n6	C20:3n6	C20:4n6	C20:5n3
1% DSW (Tank 1)	0.14	37.93	0.21	6.69	0.37	2.37	0.86	0.00	0.18	1.77	19.11	17.83
5% DSW (Tank 2)	0.13	37.51	0.14	6.90	0.36	2.18	0.81	0.00	0.04	1.92	19.05	15.87
10% DSW (Tank 3)	0.20	38.81	0.25	7.00	0.47	2.09	0.58	0.01	0.49	1.85	18.95	15.61
1% DSW (Tank 7)	0.12	38.64	0.13	6.93	0.36	2.25	0.77	0.00	0.19	1.81	20.07	16.65
5% DSW (Tank 8)	0.91	36.92	0.55	10.28	0.81	3.08	0.53	0.10	0.22	1.72	13.94	14.18
10% DSW (Tank 9)	0.20	37.57	0.23	10.09	0.48	2.79	0.56	0.06	0.29	1.80	16.30	14.78

**Table 7 plants-12-03524-t007:** Proximate biochemical compositional analysis and elemental compositional analysis of *Halymenia hawaiiana* (Trial #3), shown on the basis of dry biomass harvested from nutrient trials. Samples were taken at the end of the trial.

Tank #	Ash (%)	Lipid (%)	Protein (%)	Carbs. (%)	Uronic Acids (%)	Carbon (%)	Nitrogen (%)
0% DSW (Tank 1)	32.01	0.82	5.49	36.75	0.70	26.56	1.12
cascade from tank 1 (Tank 2)	32.16	0.83	4.23	39.83	0.65	26.96	0.86
cascade from tank 2 (Tank 3)	29.69	1.06	5.33	42.32	0.70	29.09	1.08
10% DSW (Tank 4)	32.74	0.87	15.73	25.19	0.65	27.25	3.20
cascade from tank 4 (Tank 5)	34.27	0.82	14.78	23.92	<loq	26.65	3.00
cascade from tank 5 (Tank 6)	33.56	0.81	12.28	28.44	0.65	27.10	2.50
0% DSW (Tank 7)	34.07	0.94	5.23	35.71	0.67	26.16	1.06
cascade from tank 7 (Tank 8)	31.89	0.97	4.77	38.73	<loq	26.92	0.97
10% DSW (Tank 9)	32.97	0.98	14.84	24.68	0.63	26.73	3.02

**Table 8 plants-12-03524-t008:** Carbohydrate monomeric composition of *Halymania hawaiiana* (Trial #3), shown on the basis of dry biomass harvested from nutrient trials. Samples were taken at the end of the trial. <loq (below limit of quantification) indicates where component peaks were detected, but the areas were below the calibration curve and thus no quantification was possible; N/D = not detected, Rham = rhamnose, GalN = galactosamine, Ara = arabinose, GlucN = glucosamine, Gal = galactose, Gluc = glucose, Man = mannose, Xyl = xylose, Rib = ribose.

Tank #	Rham	GalN	Ara	GlucN	Gal	Gluc	Man	Xyl	Rib
0% DSW (Tank 1)	N/D	<loq	N/D	<loq	25.75	8.93	1.46	0.33	0.29
cascade from tank 1 (Tank 2)	N/D	<loq	N/D	<loq	24.93	12.67	1.64	0.34	0.25
cascade from tank 2 (Tank 3)	<loq	<loq	N/D	<loq	20.97	19.01	1.62	0.45	0.28
10% DSW (Tank 4)	<loq	<loq	N/D	<loq	19.88	3.70	0.93	0.32	0.36
cascade from tank 4 (Tank 5)	<loq	<loq	N/D	<loq	18.76	3.48	1.05	0.29	0.34
cascade from tank 5 (Tank 6)	<loq	<loq	N/D	<loq	22.53	4.02	1.27	0.29	0.33
0% DSW (Tank 7)	<loq	<loq	N/D	<loq	23.26	10.42	1.47	0.32	0.24
cascade from tank 7 (Tank 8)	N/D	<loq	N/D	<loq	22.86	13.56	1.72	0.34	0.25
10% DSW (Tank 9)	<loq	<loq	N/D	<loq	19.50	3.59	0.99	0.30	0.29

**Table 9 plants-12-03524-t009:** Fatty acid profile of *Halymenia hawaiiana* (Trial #3), shown on the basis of total FAME content in harvested biomass from nutrient trials. Samples were taken at the end of the trial. All fatty acids detected are noted as chain length with number of unsaturated bonds, with carbon position from terminal end indicated for first unsaturation.

Tank #	C14:0	C15:0	C16:0	C16:1n7	C16:3	C16:4	C18	C18:1n9	C18:1n7	C18:2n6	C20:4n6	C20:5n3
0% DSW (Tank 1)	1.87	1.11	63.88	0.57	0.91	4.42	1.98	14.91	2.34	1.68	3.54	2.69
cascade from tank 1 (Tank 2)	1.83	0.85	65.33	0.73	0.44	5.60	2.08	11.12	1.60	1.81	4.77	3.79
cascade from tank 2 (Tank 3)	2.30	0.60	59.99	0.67	0.00	4.19	1.73	13.23	1.86	2.34	8.33	4.68
10% DSW (Tank 4)	2.06	1.49	68.71	0.68	1.18	0.96	1.92	15.74	1.71	0.40	2.53	1.97
cascade from tank 4 (Tank 5)	1.96	1.81	66.23	0.68	0.91	1.05	4.12	15.77	2.02	0.12	2.30	2.17
cascade from tank 5 (Tank 6)	1.64	1.42	63.02	0.58	1.71	1.51	3.05	18.55	2.97	0.02	2.48	2.08
0% DSW (Tank 7)	1.98	0.68	63.07	0.75	0.35	3.76	2.66	16.10	2.00	1.66	4.11	2.84
cascade from tank 7 (Tank 8)	1.98	0.44	61.04	0.73	0.01	4.84	1.29	15.54	2.29	2.12	5.59	4.08
10% DSW (Tank 9)	2.23	1.34	70.74	0.73	0.79	1.08	2.07	16.70	1.59	0.03	1.44	1.02

## Data Availability

The datasets generated during and/or analyzed during the current study are available from the corresponding author on reasonable request.

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
