# Peer review of "Tropical Red Macroalgae Cultivation with a Focus on Compositional Analysis"

_plants, 2023, doi:10.3390/plants12203524_

Round 1

Reviewer 1 Report

The manuscript “Tropical Red Macroalgae Cultivation with a Focus on Compositional Analysis, by Augyte et al. “ is an interesting article and discusses an important point. This study will help to understand the functions of polysaccharides in aquatic plants. However, several concerns need to be addressed before being published:

1.    English style and grammar need to improve. Many types of errors are found.

2.    The novelty of the study also should be presented at the end of the Intro.

3.    The first time, the scientific name should be fully written, then the abbreviation.

4.    The keywords need to improve. The following keys may be used; Polysaccharides, Carageenan, etc.

5.    The introduction needs more improvements so that the study point is presented well and needs to be more interconnected between its parts. More emphasis should be placed on the applications of polysaccharides in seaweed in the aquaculture sector.  

6.    Authors should explain the novelty of this work at the end of the introduction (before the aim of the work), and explain the commercial and scientific value of the selected species

7.    Improve the quality of Figs. 2, 3, 5, and 6.

8.    In all the manuscript parts, change “sea water ” to “seawater”

9.    Where is the “Conclusions” Section? It should provide and build on what already exists to say something about the relevance of the findings to current research, future research, and the implications of the study.

Extensive editing of English language required

Reviewer 2 Report

This study examined the biomass and metabolic composition of two red algae using different concentrations of deep ocean water. The background and aim of this study are clear and the description of results is accurate and easy to understand. However, the culture method and system of some experiments are not clear for me and some of the results are not discussed well. I think these issues should be improved before publication. The detailed comments are listed below.

 Abstract: The latter half of abstract is occupied by more general comments and the results of this study are not fully summarized.

 Introduction: All abbreviations should be spelled out when they are mentioned at the first time.

 Results, Trial #1: Please explain the reason this experiment was performed only for Agardhiella.

Results, Trial #2 and #3: It is helpful to show a schematic depiction for an easy understanding of this culture system. I do not understand why DSW was supplied again to tank 9.

 Figs 4 and 7: Which tank of these materials? Is there no difference of thallus condition between the tanks?

 Fig. 6: The explanation is only for week 0 and 1.

Table 3: Please explain why there is no table for average yields of Trial #2. Which tank of this average yields?

Results, p8: "Only the measurable ... present in the biomass." "... which is consistent with ... energy-storage functions." These sentences should be moved to Discussion.

Tables 4-9: I think the detailed data in these tables are dispensable to understand the essence of this study and can be inserted as supplements. Instead, the statistical test should be performed to focus on the significant difference of each component between the conditions.

Discussion p.10: "... the macroalgae can maintain high growth rates of over 26%". But the growth rate drops down after the trimming. Why?

Discussion p.11: I do not know the unit "T acre-1". 

Discussion p.11: "T m-2 yr-1" '-2' should be superscript. 

Materials and Methods p.12: I wonder if the seawater temperature is constant throughout the tanks. The downstream tank must be more affected by air temperature and sun light. If so, this different temperature between the tanks may affect the algal growth. 

Materials and Methods p.12: "... tropical red algae Halymenia hawaiiana and Agardhiella subulata ..."  The order of the two species should be the same throughout the text. Please explain the date and place the materials were collected for each trial and how the materials were maintained before the experiments. 

Materials and Methods p.12: "... clearing away any fouling on a weekly basis." How was the fouling condition? Fouling issue is not always ignorable and it is important to consider the time and cost to remove epibionts. 

Materials and Methods p.12: "Because the trimming event ... 5 weeks as period B." It is necessary to show this explanation in the table caption. 

Materials and Methods p.13: "... average temperature 23ËšC," DSW is colder than SSW, so the temperature must vary between the different DSW concentrations.

Round 2

Reviewer 1 Report

Accept in present form

Reviewer 2 Report

This manuscript was revised accordingly.